# Evidence for the utility of quantum computing before fault tolerance

Youngseok Kim[1,6 ✉], Andrew Eddins[2,6 ✉], Sajant Anand[3], Ken Xuan Wei[1], Ewout van den Berg[1], Sami Rosenblatt[1], Hasan Nayfeh[1], Yantao Wu[3,4], Michael Zaletel[3,5], Kristan Temme[1] & Abhinav Kandala[1 ✉]

Quantum computing promises to offer substantial speed-ups over its classical counterpart for certain problems. However, the greatest impediment to realizing its full potential is noise that is inherent to these systems. The widely accepted solution to this challenge is the implementation of fault-tolerant quantum circuits, which is out of reach for current processors. Here we report experiments on a noisy 127-qubit processor and demonstrate the measurement of accurate expectation values for circuit volumes at a scale beyond brute-force classical computation. We argue that this represents evidence for the utility of quantum computing in a pre-fault-tolerant era. These experimental results are enabled by advances in the coherence and calibration of a superconducting processor at this scale and the ability to characterize[1] and controllably manipulate noise across such a large device. We establish the accuracy of the measured expectation values by comparing them with the output of exactly verifiable circuits. In the regime of strong entanglement, the quantum computer provides correct results for which leading classical approximations such as pure-state-based 1D (matrix product states, MPS) and 2D (isometric tensor network states, isoTNS) tensor network methods[2,3] break down. These experiments demonstrate a foundational tool for the realization of near-term quantum applications[4,5].

It is almost universally accepted that advanced quantum algorithms such as factoring[6] or phase estimation[7] will require quantum error correction. However, it is acutely debated whether processors available at present can be made sufficiently reliable to run other, shorter-depth quantum circuits at a scale that could provide an advantage for practical problems. At this point, the conventional expectation is that the implementation of even simple quantum circuits with the potential to exceed classical capabilities will have to wait until more advanced, fault-tolerant processors arrive. Despite the tremendous progress of quantum hardware in recent years, simple fidelity bounds[8] support this bleak forecast; one estimates that a quantum circuit 100 qubits wide by 100 gate-layers deep executed with 0.1% gate error yields a state fidelity less than $5 \times 10^{-4}$. Nonetheless, the question remains whether properties of the ideal state can be accessed even with such low fidelities. The error-mitigation[9,10] approach to near-term quantum advantage on noisy devices exactly addresses this question, that is, that one can produce accurate expectation values from several different runs of the noisy quantum circuit using classical post-processing.

Quantum advantage can be approached in two steps: first, by demonstrating the ability of existing devices to perform accurate computations at a scale that lies beyond brute-force classical simulation, and second by finding problems with associated quantum circuits that derive an advantage from these devices. Here we focus on taking the

first step and do not aim to implement quantum circuits for problems with proven speed-ups.

We use a superconducting quantum processor with 127 qubits to run quantum circuits with up to 60 layers of two-qubit gates, a total of 2,880 CNOT gates. General quantum circuits of this size lie beyond what is feasible with brute-force classical methods. We thus first focus on specific test cases of the circuits permitting exact classical verification of the measured expectation values. We then turn to circuit regimes and observables in which classical simulation becomes challenging and compare with results from state-of-the-art approximate classical methods.

Our benchmark circuit is the Trotterized time evolution of a 2D transverse-field Ising model, sharing the topology of the qubit processor (Fig. 1a). The Ising model appears extensively across several areas in physics and has found creative extensions in recent simulations exploring quantum many-body phenomena, such as time crystals[11,12], quantum scars[13] and Majorana edge modes[14]. As a test of utility of quantum computation, however, the time evolution of the 2D transverse-field Ising model is most relevant in the limit of large entanglement growth in which scalable classical approximations struggle.

In particular, we consider time dynamics of the Hamiltonian,

$$H = -J \sum_{\langle i,j \rangle} Z_i Z_j + h \sum_i X_i, \qquad (1)$$

[1]IBM Quantum, IBM Thomas J. Watson Research Center, Yorktown Heights, NY, USA. [2]IBM Quantum, IBM Research - Cambridge, Cambridge, MA, USA. [3]Department of Physics, University of California, Berkeley, Berkeley, CA, USA. [4]RIKEN iTHEMS, Wako, Japan. [5]Materials Sciences Division, Lawrence Berkeley National Laboratory, Berkeley, CA, USA. [6]These authors contributed equally: Youngseok Kim, Andrew Eddins. ✉e-mail: youngseok.kim1@ibm.com; aeddins@ibm.com; akandala@us.ibm.com

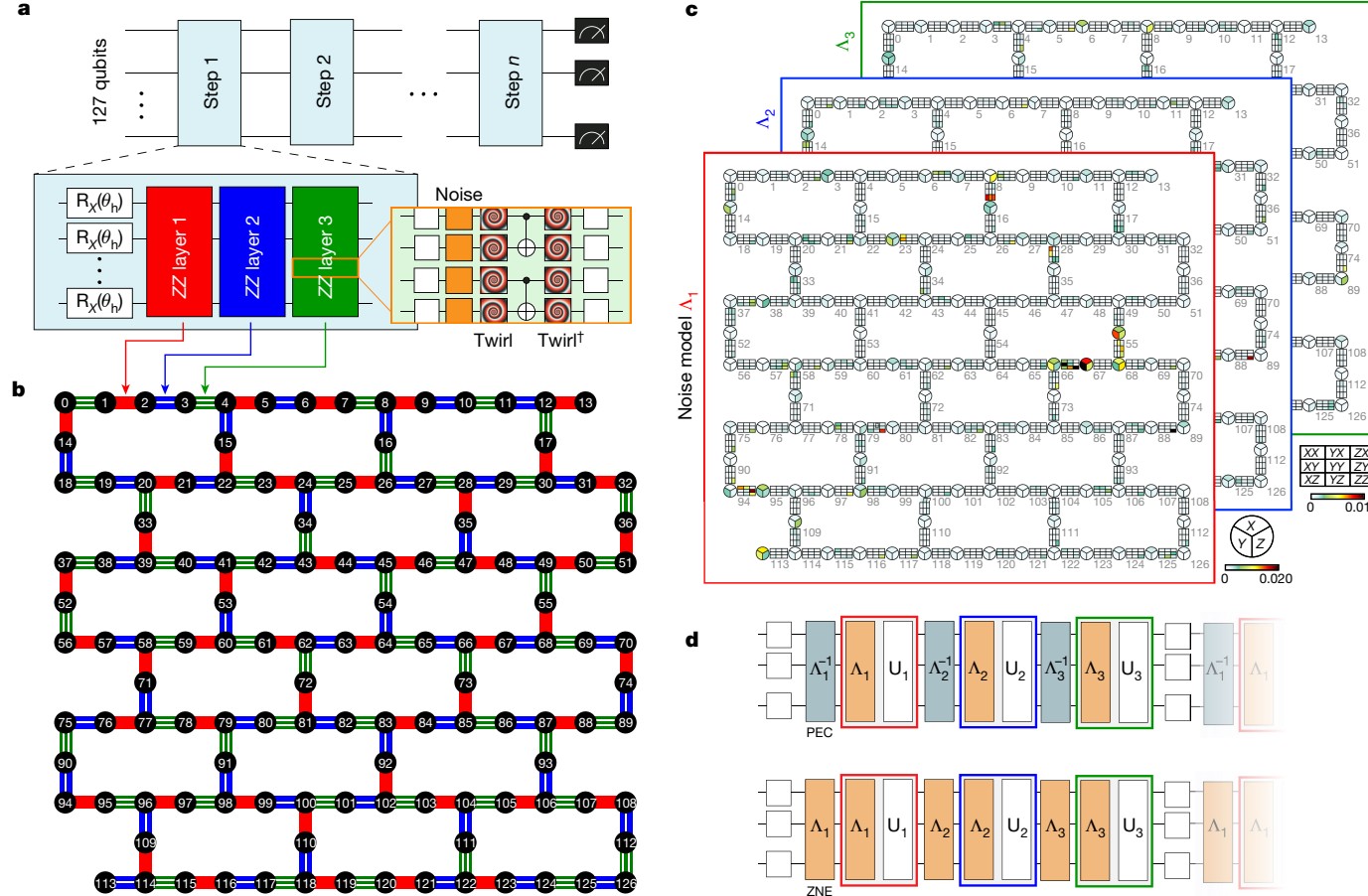

**Fig. 1 | Noise characterization and scaling for 127-qubit Trotterized time-evolution circuits. a**, Each Trotter step of the Ising simulation includes single-qubit $X$ and two-qubit $ZZ$ rotations. Random Pauli gates are inserted to twirl (spirals) and controllably scale the noise of each CNOT layer. The dagger indicates conjugation by the ideal layer. **b**, Three depth-1 layers of CNOT gates suffice to realize interactions between all neighbour pairs on ibm_kyiv.

**c**, Characterization experiments efficiently learn the local Pauli error rates $\lambda_{l,i}$ (colour scales) comprising the overall Pauli channel $\Lambda_l$ associated with the $l$th twirled CNOT layer. (Figure expanded in Supplementary Information IV.A). **d**, Pauli errors inserted at proportional rates can be used to either cancel (PEC) or amplify (ZNE) the intrinsic noise.

in which $J > 0$ is the coupling of nearest-neighbour spins with $i < j$ and $h$ is the global transverse field. Spin dynamics from an initial state can be simulated by means of first-order Trotter decomposition of the time-evolution operator,

$$e^{-iH_{ZZ}\delta t} = \prod_{\langle i,j \rangle} e^{iJ\delta t Z_i Z_j} = \prod_{\langle i,j \rangle} R_{Z_i Z_j}(-2J\delta t)$$
$$e^{-iH_X \delta t} = \prod_i e^{-ih\delta t X_i} = \prod_i R_{X_i}(2h\delta t),$$
(2)

in which the evolution time $T$ is discretized into $T/\delta t$ Trotter steps and $R_{Z_i Z_j}(\theta_J)$ and $R_{X_i}(\theta_h)$ are $ZZ$ and $X$ rotation gates, respectively. We are not concerned with the model error owing to Trotterization and thus take the Trotterized circuit as ideal for any classical comparison. For experimental simplicity, we focus on the case $\theta_J = -2J\delta t = -\pi/2$ such that the $ZZ$ rotation requires only one CNOT,

$$R_{ZZ}\left(\frac{-\pi}{2}\right) = \begin{array}{c} S^\dagger \\ S^\dagger \quad \sqrt{Y} \quad \oplus \quad \sqrt{Y}^\dagger \end{array},$$

where the equality holds up to a global phase. In the resulting circuit (Fig. 1a), each Trotter step amounts to a layer of single-qubit rotations, $R_X(\theta_h)$, followed by commuting layers of parallelized two-qubit rotations, $R_{ZZ}(\theta_J)$.

For the experimental implementation, we primarily used the IBM Eagle processor ibm_kyiv, composed of 127 fixed-frequency transmon

qubits[15] with heavy-hex connectivity and median $T_1$ and $T_2$ times of 288 μs and 127 μs, respectively. These coherence times are unprecedented for superconducting processors of this scale and allow the circuit depths accessed in this work. The two-qubit CNOT gates between neighbours are realized by calibrating the cross-resonance interaction[16]. As each qubit has at most three neighbours, all $ZZ$ interactions can be performed in three layers of parallelized CNOT gates (Fig. 1b). The CNOT gates within each layer are calibrated for optimal simultaneous operation (see Methods for more details).

We now see that these hardware performance improvements enable even larger problems to be successfully executed with error mitigation, in comparison with recent work[1,17] on this platform. Probabilistic error cancellation (PEC)[9] has been shown[1] to be very effective at providing unbiased estimates of observables. In PEC, a representative noise model is learned and effectively inverted by sampling from a distribution of noisy circuits related to the learned model. Yet, for the current error rates on our device, the sampling overhead for the circuit volumes considered in this work remains restrictive, as discussed further below.

We therefore turn to zero-noise extrapolation (ZNE)[9,10,17,18], which provides a biased estimator at a potentially much lower sampling cost. ZNE is either a polynomial[9,10] or exponential[19] extrapolation method for noisy expectation values as a function of a noise parameter. This requires the controlled amplification of the intrinsic hardware noise by a known gain factor $G$ to extrapolate to the ideal $G = 0$ result. ZNE has been widely adopted in part because noise-amplification schemes

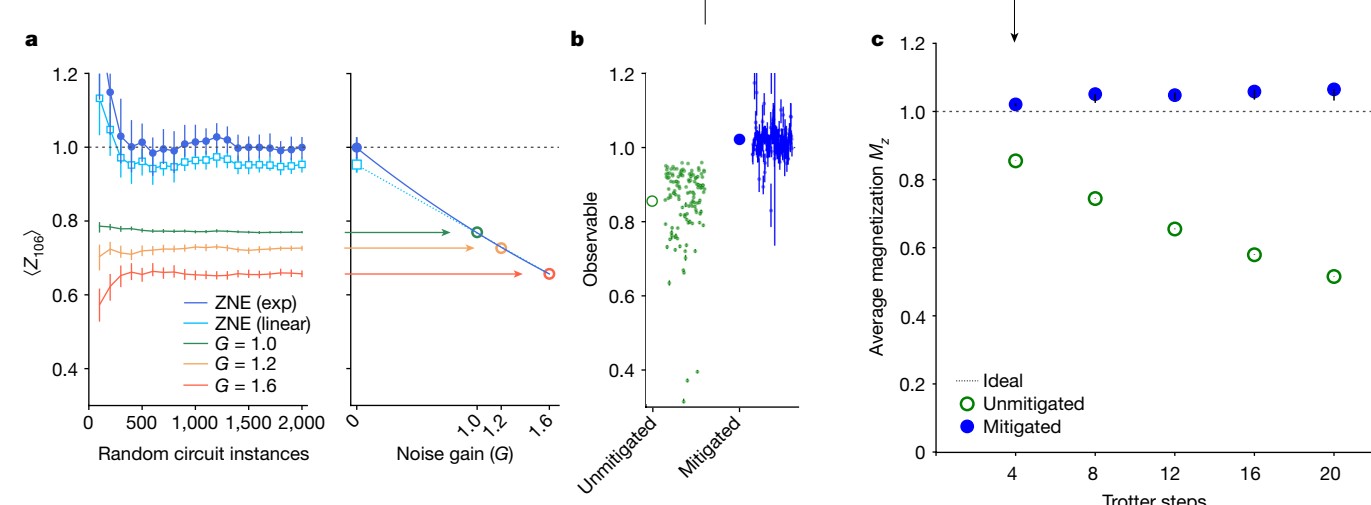

**Fig. 2 | Zero-noise extrapolation with probabilistic error amplification.** Mitigated expectation values from Trotter circuits at the Clifford condition $\theta_h = 0$. **a**, Convergence of unmitigated ($G = 1$), noise-amplified ($G > 1$) and noise-mitigated (ZNE) estimates of $\langle Z_{106} \rangle$ after four Trotter steps. In all panels, error bars indicate 68% confidence intervals obtained by means of percentile bootstrap. Exponential extrapolation (exp, dark blue) tends to outperform linear extrapolation (linear, light blue) when differences between the converged estimates of $\langle Z_{106} \rangle_{G \neq 0}$ are well resolved. **b**, Magnetization (large markers) is computed as the mean of the individual estimates of $\langle Z_q \rangle$ for all qubits (small markers). **c**, As circuit depth is increased, unmitigated estimates of $M_z$ decay monotonically from the ideal value of 1. ZNE greatly improves the estimates even after 20 Trotter steps (see Supplementary Information II for ZNE details).

based on pulse stretching[9,17,18] or subcircuit repetition[20–22] have circumvented the need for precise noise learning, while relying on simplistic assumptions about the device noise. More precise noise amplification can, however, enable substantial reductions in the bias of the extrapolated estimator, as we demonstrate here.

The sparse Pauli–Lindblad noise model proposed in ref. 1 turns out to be especially well suited for noise shaping in ZNE. The model takes the form $e^{\mathcal{L}}$, in which $\mathcal{L}$ is a Lindbladian comprising Pauli jump operators $P_i$ weighted by rates $\lambda_i$. It was shown in ref. 1 that restricting to jump operators acting on local pairs of qubits yields a sparse noise model that can be efficiently learned for many qubits and that accurately captures the noise associated with layers of two-qubit Clifford gates, including crosstalk, when combined with random Pauli twirls[23,24]. The noisy layer of gates is modelled as a set of ideal gates preceded by some noise channel $\Lambda$. Thus, applying $\Lambda^\alpha$ before the noisy layer produces an overall noise channel $\Lambda^G$ with gain $G = \alpha + 1$. Given the exponential form of the Pauli–Lindblad noise model, the map $e^{\alpha \mathcal{L}}$ is obtained by simply multiplying the Pauli rates $\lambda_i$ by $\alpha$. The resulting Pauli map can be sampled to obtain appropriate circuit instances; for $\alpha \geq 0$, the map is a Pauli channel that can be sampled directly, whereas for $\alpha < 0$, quasi-probabilistic sampling is needed with sampling overhead $\gamma^{-2\alpha}$ for some model-specific $\gamma$. In PEC, we choose $\alpha = -1$ to obtain an overall zero-gain noise level. In ZNE, we instead amplify the noise[10,25–27] to different gain levels and estimate the zero-noise limit using extrapolation. For practical applications, we need to consider the stability of the learned noise model over time (Supplementary Information III.A), for instance, owing to qubit interactions with fluctuating microscopic defects known as two-level systems[28].

Clifford circuits serve as useful benchmarks of estimates produced by error mitigation, as they can be efficiently simulated classically[29]. Notably, the entire Ising Trotter circuit becomes Clifford when $\theta_h$ is chosen to be a multiple of $\pi/2$. As a first example, we therefore set the transverse field to zero ($R_x(0) = I$) and evolve the initial state $|0\rangle^{\otimes 127}$ (Fig. 1a). The CNOT gates nominally leave this state unchanged, so the ideal weight-1 observables $Z_q$ all have expectation value 1; owing to the Pauli twirling of each layer, the bare CNOTs do affect the state. For each Trotter experiment, we first characterized the noise models $\Lambda_l$ for the

three Pauli-twirled CNOT layers (Fig. 1c) and then used these models to implement Trotter circuits with noise gain levels $G \in \{1, 1.2, 1.6\}$. Figure 2a illustrates the estimation of $\langle Z_{106} \rangle$ after four Trotter steps (12 CNOT layers). For each $G$, we generated 2,000 circuit instances in which, before each layer $l$, we have inserted products of one-qubit and two-qubit Pauli errors $i$ from $\mathcal{L}$ drawn with probabilities $p_{l,i} = (1 - e^{-2(G-1)\lambda_{l,i}})/2$ and executed each instance 64 times, totalling 384,000 executions. As more circuit instances are accumulated, the estimates of $\langle Z_{106} \rangle_G$, corresponding to the different gains $G$, converge to distinct values. The different estimates are then fit by an extrapolating function in $G$ to estimate the ideal value $\langle Z_{106} \rangle_0$. The results in Fig. 2a highlight the reduced bias from exponential extrapolation[19] in comparison with linear extrapolation. That said, exponential extrapolation can exhibit instabilities, for instance, when expectation values are unresolvably close to zero, and—in such cases—we iteratively downgrade the extrapolation model complexity (see Supplementary Information II.B). The procedure outlined in Fig. 2a was applied to the measurement results from each qubit $q$ to estimate all $N = 127$ Pauli expectations $\langle Z_q \rangle_0$. The variation in the unmitigated and mitigated observables in Fig. 2b is indicative of the non-uniformity in the error rates across the entire processor. We report the global magnetization along $\hat{z}$, $M_z = \sum_q \langle Z_q \rangle / N$, for increasing depth in Fig. 2c. Although the unmitigated result shows a gradual decay from 1 with an increasing deviation for deeper circuits, ZNE greatly improves agreement, albeit with a small bias, with the ideal value even out to 20 Trotter steps, or 60 CNOT depth. Notably, the number of samples used here is much smaller than an estimate of the sampling overhead that would be needed in a naive PEC implementation (see Supplementary Information IV.B). In principle, this disparity may be greatly reduced by more advanced PEC implementations using light-cone tracing[30] or by improvements in hardware error rates. As future hardware and software developments bring down sampling costs, PEC may be preferred when affordable to avoid the potentially biased nature of ZNE.

Next, we test the efficacy of our methods for non-Clifford circuits and the Clifford $\theta_h = \pi/2$ point, with non-trivial entangling dynamics compared with the identity-equivalent circuits discussed in Fig. 2. The non-Clifford circuits are of particular importance to test,

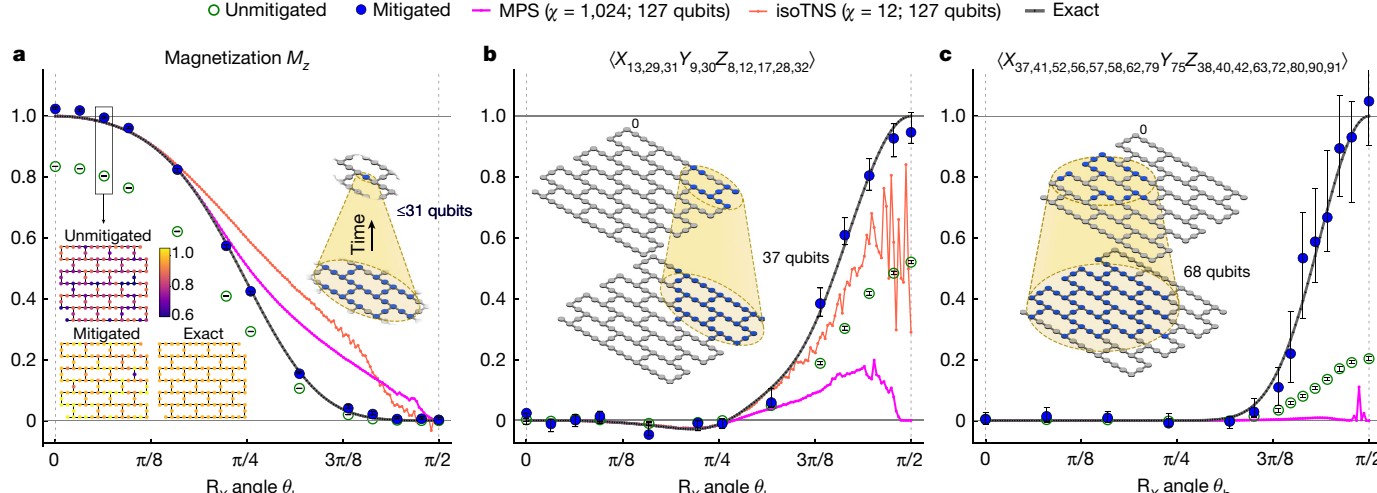

**Fig. 3 | Classically verifiable expectation values from 127-qubit, depth-15 Clifford and non-Clifford circuits.** Expectation value estimates for $\theta_h$ sweeps at a fixed depth of five Trotter steps for the circuit in Fig. 1a. The considered circuits are non-Clifford except at $\theta_h = 0, \pi/2$. Light-cone and depth reductions of respective circuits enable exact classical simulation of the observables for all $\theta_h$. For all three plotted quantities (panel titles), mitigated experimental results (blue) closely track the exact behaviour (grey). In all panels, error bars indicate 68% confidence intervals obtained by means of percentile bootstrap. The weight-10 and weight-17 observables in **b** and **c** are stabilizers of the circuit at $\theta_h = \pi/2$ with respective eigenvalues +1 and −1; all values in **c** have been negated for visual simplicity. The lower inset in **a** depicts variation of $\langle Z_q \rangle$ at $\theta_h = 0.2$ across the device before and after mitigation and compares with exact results.

Upper insets in all panels illustrate causal light cones, indicating in blue the final qubits measured (top) and the nominal set of initial qubits that can influence the state of the final qubits (bottom). $M_z$ also depends on 126 other cones besides the example shown. Although in all panels exact results are obtained from simulations of only causal qubits, we include tensor network simulations of all 127 qubits (MPS, isoTNS) to help gauge the domain of validity for those techniques, as discussed in the main text. isoTNS results for the weight-17 operator in **c** are not accessible with current methods (see Supplementary Information VI). All experiments were carried out for $G = 1, 1.2, 1.6$ and extrapolated as in Supplementary Information II.B. For each $G$, we generated 1,800–2,000 random circuit instances for **a** and **b** and 2,500–3,000 instances for **c**.

as the validity of exponential extrapolation is no longer guaranteed (see Supplementary Information V and ref. 31). We restrict the circuit depth to five Trotter steps (15 CNOT layers) and judiciously choose observables that are exactly verifiable. Figure 3 shows the results as $\theta_h$ is swept between 0 and $\pi/2$ for three such observables of increasing weight. Figure 3a shows $M_z$ as before, an average of weight-1 $\langle Z \rangle$ observables, whereas Fig. 3b,c show weight-10 and weight-17 observables. The latter operators are stabilizers of the Clifford circuit at $\theta_h = \pi/2$, obtained by evolution of the initial stabilizers $Z_{13}$ and $Z_{58}$, respectively, of $|0\rangle^{\otimes 127}$ for five Trotter steps, ensuring non-vanishing expectation values in the strongly entangling regime of particular interest. Although the entire 127-qubit circuit is executed experimentally, light-cone and depth-reduced (LCDR) circuits enable brute-force classical simulation of the magnetization and weight-10 operator at this depth (see Supplementary Information VII). Over the full extent of the $\theta_h$ sweep, the error-mitigated observables show good agreement with the exact evolution (see Fig. 3a,b). However, for the weight-17 operator, the light cone expands to 68 qubits, a scale beyond brute-force classical simulation, so we turn to tensor network methods.

Tensor networks have been widely used to approximate and compress quantum state vectors that arise in the study of the low-energy eigenstates of and time evolution by local Hamiltonians[2,32,33] and, more recently, have been successfully used to simulate low-depth noisy quantum circuits[34–36]. Simulation accuracy can be improved by increasing the bond dimension $\chi$, which constrains the amount of entanglement of the represented quantum state, at a computational cost scaling polynomially with $\chi$. As entanglement (bond dimension) of a generic state grows linearly (exponentially) with time evolution until it saturates the volume law, deep quantum circuits are inherently difficult for tensor networks[37]. We consider both quasi-1D matrix product states (MPS) and 2D isometric tensor network states (isoTNS)[3] that have $\mathcal{O}(\chi^3)$ and $\mathcal{O}(\chi^7)$ scaling of time-evolution complexity, respectively. Details of both methods and their strengths are provided in Methods

and Supplementary Information VI. Specifically for the case of the weight-17 operator shown in Fig. 3c, we find that an MPS simulation of the LCDR circuit at $\chi = 2,048$ is sufficient to obtain the exact evolution (see Supplementary Information VIII). The larger causal cone of the weight-17 observable results in an experimental signal that is weaker compared with that of the weight-10 observable; nevertheless, mitigation still yields good agreement with the exact trace. This comparison suggests that the domain of experimental accuracy could extend beyond the scale of exact classical simulation.

We expect that these experiments will eventually extend to circuit volumes and observables in which such light-cone and depth reductions are no longer important. Therefore, we also study the performance of MPS and isoTNS for the full 127-qubit circuit executed in Fig. 3, at respective bond dimensions of $\chi = 1,024$ and $\chi = 12$, which are primarily limited by memory requirements. Figure 3 shows that the tensor network methods struggle with increasing $\theta_h$, losing both accuracy and continuity near the verifiable Clifford point $\theta_h = \pi/2$. This breakdown can be understood in terms of entanglement properties of the state. The stabilizer state produced by the circuit at $\theta_h = \pi/2$ has an exactly flat bipartite entanglement spectrum, found from a Schmidt decomposition of a 1D ordering of the qubits. Thus, truncating states with small Schmidt weight—the basis of all tensor network algorithms—is not justified. However, as exact tensor network representations generically require bond dimension exponential in circuit depth, truncation is necessary for tractable numerical simulations.

Finally, in Fig. 4, we stretch our experiments to regimes in which the exact solution is not available with the classical methods considered here. The first example (Fig. 4a) is similar to Fig. 3c but with a further final layer of single-qubit Pauli rotations that interrupt the circuit-depth reduction that previously enabled exact verification for any $\theta_h$ (see Supplementary Information VII). At the verifiable Clifford point $\theta_h = \pi/2$, the mitigated results agree again with the ideal value, whereas the $\chi = 3,072$ MPS simulation of the 68-qubit LCDR circuit markedly fails

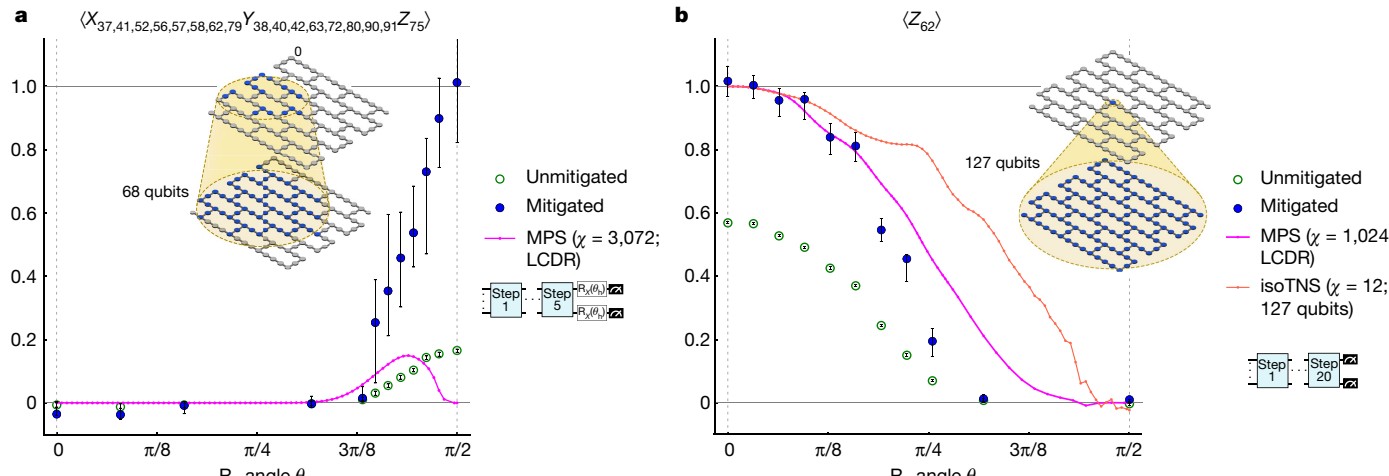

**Fig. 4 | Estimating expectation values beyond exact verification.** Plot markers, confidence intervals and causal light cones appear as defined in Fig. 3. **a**, Estimates of a weight-17 observable (panel title) after five Trotter steps for several values of $\theta_h$. The circuit is similar to that in Fig. 3c but with further single-qubit rotations at the end. This effectively simulates the time evolution of the spins after Trotter step six by using the same number of two-qubit gates used for Trotter step five. As in Fig. 3c, the observable is a stabilizer at $\theta_h = \pi/2$ with eigenvalue −1, so we negate the $y$ axis for visual simplicity. Optimization of the MPS simulation by including only qubits and gates in the causal light cone enables a higher bond dimension ($\chi = 3,072$), but the simulation still fails to approach −1 (+1 in negated $y$ axis) at $\theta_h = \pi/2$. **b**, Estimates of the single-site magnetization $\langle Z_{62} \rangle$ after 20 Trotter steps for several values of $\theta_h$. The MPS simulation is light-cone-optimized and performed with bond dimension $\chi = 1,024$, whereas the isoTNS simulation ($\chi = 12$) includes the gates outside the light cone. The experiments were carried out with $G = 1, 1.3, 1.6$ for **a** and $G = 1, 1.2, 1.6$ for **b**, and extrapolated as in Supplementary Information II.B. For each $G$, we generated 2,000–3,200 random circuit instances for **a** and 1,700–2,400 instances for **b**.

in the strongly entangling regime of interest. Although $\chi = 2,048$ was sufficient for exact simulation of the weight-17 operator in Fig. 3c, an MPS bond dimension of 32,768 would be needed for exact simulation of this modified circuit and operator with $\theta_h = \pi/2$.

As a final example, we extend the circuit depth to 20 Trotter steps (60 CNOT layers) and estimate the $\theta_h$ dependence of a weight-1 observable, $\langle Z_{62} \rangle$, in Fig. 4b, in which the causal cone extends over the entire device. Given the non-uniformity of device performance, also seen in the spread of single-site observables in Fig. 2b, we choose an observable that obtains the expected result $\langle Z_{62} \rangle \approx 1$ at the verifiable $\theta_h = 0$ point. Despite the greater depth, the MPS simulations of the LCDR circuit agree well with the experiment in the weakly entangling regime of small $\theta_h$. Although deviations from the experimental trace emerge with increasing $\theta_h$, we note that the MPS simulations slowly move in the direction of the experimental data with increasing $\chi$ (see Supplementary Information X) and that the bond dimension needed to exactly represent the stabilizer state and its evolution to depth 20 at $\theta_h = \pi/2$ is $7.2 \times 10^{16}$, 13 orders of magnitude larger than what we considered (see Supplementary Information VIII). For reference, as the memory required to store an MPS scales as $\mathcal{O}(\chi^2)$, already a bond dimension of $\chi = 1 \times 10^8$ would require 400 PB, independent of any runtime considerations. Furthermore, full-state tensor network simulations are already unable to capture the dynamics at the exactly verifiable five-step circuit in Fig. 3a. We also note that, given the large unmitigated signal, there may be opportunity to study time evolution at even larger depths on the current device.

For execution times, the tensor network simulations in Fig. 4 were run on a 64-core, 2.45-GHz processor with 128 GB of memory, in which the run time to access an individual data point at fixed $\theta_h$ was 8 h for Fig. 4a and 30 h for Fig. 4b. The corresponding quantum wall-clock run time was approximately 4 h for Fig. 4a and 9.5 h for Fig. 4b, but this is also far from a fundamental limit, being at present dominated by classical processing delays that stand to be largely eliminated through conceptually straightforward optimizations. Indeed, the estimated device run time for the error-mitigated expectation values using 614,400 samples (2,400 circuit instances for each gain factor and readout error

mitigation, with 64 shots per instance) at a conservative sampling rate of 2 kHz is only 5 min 7 s, which can be even further reduced by optimization of qubit reset speeds. On the other hand, the classical simulations may also be improved by methods besides the pure-state tensor networks considered here, such as Heisenberg operator evolution methods, which have recently been applied to non-Clifford simulations[38]. Another approach is to numerically emulate the ZNE used experimentally. For example, it was recently argued that the finite-$\chi$ truncation error introduced by tensor-product compression mimics experimental gate errors[34]. It would thus be natural to develop a theory for extrapolating tensor network state expectation values in the bond dimension $\chi$ for time evolution, as has been done in the case of ground-state search[39]. Alternatively, one can more directly emulate ZNE by introducing artificial dissipation into the dynamics engineered so that the resulting mixed-state evolution has reduced tensor-product bond dimension, as—for example—in dissipation-assisted operator evolution[40], and extrapolate results with respect to the strength of the dissipation. Although such methods[40,41] can successfully capture the long-time dynamics of the low-weight observables of a 1D spin chain, their applicability to high-weight observables in 2D at intermediate times is not clear—particularly as these methods are explicitly constructed to truncate complex operators.

The observation that a noisy quantum processor, even before the advent of fault-tolerant quantum computing, produces reliable expectation values at a scale beyond 100 qubits and non-trivial circuit depth leads to the conclusion that there is indeed merit to pursuing research towards deriving a practical computational advantage from noise-limited quantum circuits. Over recent years, substantial research effort has been directed to develop and demonstrate candidate heuristic quantum algorithms[5] that use noise-limited quantum circuits to estimate expectation values. We have now reached reliability at a scale for which one will be able to verify proposals and explore new approaches to determine which can provide utility beyond classical approximation methods. At the same time, these results will motivate and help advance classical approximation methods as both approaches serve as valuable benchmarks of one another. However, even with improved

classical methods, impending order-of-magnitude improvements in gate fidelities[42] and speed of superconducting quantum systems will drive substantial enhancements in accessible circuit volumes and paint an increasingly bright picture of the utility of noisy quantum computers.

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

# Article

## Methods

### Device calibration

The speed of cross-resonance-based CNOT gates is dependent on the qubit–qubit detuning and, typically, gate speeds across the device are chosen independently to minimize individual gate errors[43]. This leads to a large spread in CNOT times across the device. Noting that the speed of each parallelized CNOT layer is limited by the slowest gate in the layer, we develop a new tune-up scheme for large-scale processor calibration that optimizes the layer rather than the individual gates. First, the control and target qubits are assigned to each gate layer to reduce crosstalk and leakage from transmon-frequency collisions. The slowest gate in each layer then has its duration carefully optimized. Finally, all gates in the layer are fixed to this duration and calibrated simultaneously with error-amplification sequences[44]. Compared with independently calibrated gates, the layer duration is unchanged, but gates are slower with lower drive amplitudes, reducing any leakage arising from multi-photon transitions. The simultaneous calibration also ensures that the gates are calibrated as they are implemented in the circuit.

### Noise model

Throughout this work, we amplify gate noise by means of a learned noise model. For this model, following ref. 1, a general Pauli channel is approximated by $\Lambda(\rho) = \exp[\mathcal{L}](\rho)$ with a sparse Pauli–Lindblad generator

$$\mathcal{L}(\rho) = \sum_i \lambda_i (P_i \rho P_i^\dagger - \rho).$$

Here the jump operators are chosen to be Pauli operators $P_i$ with $P_i^\dagger P_i = I$ and the model is parameterized by the non-negative coefficients $\lambda_i$. This model can be rewritten as

$$\Lambda(\rho) = \underset{i}{\bigcirc}(w_i \cdot + (1 - w_i)P_i \cdot P_i^\dagger)(\rho),$$

in which $w_i = (1 + e^{-2\lambda_i})/2$ and $\bigcirc_{i=1}^n O_i(\cdot) = (O_n \circ O_{n-1} \circ \cdots \circ O_1)$ represents the composition of operators and $O(\cdot)(\rho) = O(\rho)$. In other words, we can express $\Lambda(\rho)$ as a composition of simple Pauli maps. For physical noise channels, in which all $\lambda_i \geq 0$, the composition consists of simply Pauli channels. By allowing non-zero coefficients $\lambda_i$ only for Pauli terms $P_i$ whose support corresponds to a single qubit or a pair of connected qubits, we obtain a sparse noise model that can be efficiently learned and that, despite its simplicity, is able to capture crosstalk errors[1]. It is readily seen that $\exp[\alpha\mathcal{L}]$ is obtained by scaling all $\lambda_i$ by $\alpha$. For $\alpha \geq 0$, the resulting noise model is a composition of Pauli channels. Samples from this channel can be obtained by independently sampling $P_i$ with probability $1 - w_i$ for each of the simple channels and multiplying the results. For $\alpha < 0$, the resulting coefficients $1 - w_i$ are generally negative, leading to a non-physical noise map. Sampling in that case can still be done, albeit in a quasi-probabilistic manner. Doing so results in a sampling overhead of $\gamma^2$, in which $\gamma = \exp(\sum_i 2\lambda_i)$.

### Brute-force simulations

The simplest, most accurate and most limited method is simulation of a collection of the state of $M$ qubits as a dense vector of $2^M$ complex coefficients. All unitary gates, irrespective of locality, can be applied directly to the vector. Expectation values are found by vector–matrix–vector product of the conjugated state, operator and state. We use this approach for simulations up to 30 qubits.

### Tensor network methods

For circuits of more than 30 qubits, we used 1D and 2D tensor network state methods[45]. For a quantum state on $M$ qubits, tensor network methods approximate the $2^M$ complex coefficients for the wavefunction amplitude as a network of contracted tensors containing $\mathcal{O}(M\chi^p)$ coefficients, in which $p$ is an integer depending on the method. Here we consider MPS[2,32,33] with $p = 2$ and isoTNS[3] with $p = 4$. MPS represent a quantum state as a network of rank-3 tensors that, when contracted or multiplied together, give an approximation to the wavefunction amplitude for each basis state. isoTNS are a restriction of projected entangled pair states, a 2D generalization of MPS to square lattices in which the network consists of rank-5 tensors. The accuracy and computational cost of both MPS and isoTNS depend on the bond dimension $\chi$. MPS methods have the advantage of well-developed algorithms, yet suffer from fundamental limitations of using a 1D method to simulate a 2D system. isoTNS methods, on the other hand, are inherently 2D methods but suffer from unavoidable sources of error not present for MPS, though these can be systematically reduced with increasing bond dimension.

## Data availability

The datasets generated and analysed during this study are available at https://doi.org/10.6084/m9.figshare.22500355.

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

**Acknowledgements** This work would not have been possible without the contributions of the entire IBM Quantum team, including: K. Aarhus, B. Abdo, D. W. Abraham, E. Adams, G. Aleksandrowicz, D. Alevras, T. Alexander, A. Alexopoulos, H. Alghassi, L. Ament, M. Amico, A. Anderson, M. Aney, S. A. S. Antezana, J. Apuzzo, E. Arbel, E. Arellano, V. Arena, T. Armon, M. Arthur, P. K. Austel, U. Bacher, S. Bangsaruntip, Z. Barabás, v. barbosa, S. V. Barron, G. Bauer, J. Bauer, M. Beck, M. Beckley, S. W. Bedell, L. Bello, Y. Ben-Haim, G. Bennett, L. A. Berge, M. Bernagozzi, J. Betke, L. S. Bishop, J. Blair, A. A. Blanco, S. R. Blanks, D. F. Bogorin, R. Bonam, M. Boraas, P. Bosavage, Y. Bosch, S. Bravyi, M. Brink, B. J. Brown, J. S. Broz, J. Bruley, D. Bryant, M. Buehler, M. Byers, M. T. Byrnes, E. Bäumer, C. Cabral Jr., S. Cairns, J. Calderón, L. Carata-Dejoianu, M. Carlucci, S. Carnevale, A. Carniol, S. Carri, M. S. Carroll, A. Carter, E. C. Castañeda, R. Chadwick, M. E. V. Chan, R. Cheek, V. Chiang, R. Chiarella, M. Chido, S. Choi, R. Cholarajan, A. Choquette, J. M. Chow, T. A. Christensen, B. Christenson, C. Christianson, W. Chung, C. Chung, M. Clausen, C. Codella, H. Colquhoun Jr., R. Conti, M. Copel, A. D. Corcoles, H. Cortes, J. M. Cotte, R. Cremins, A. Cross, A. Cross, G. Crossman, J. Cruz-Benito, S. Czaplewski, P. E. Dahlen, H. P. Dang, K. Dang, J. Dangler, M. P. David, A. R. Davila, G. Dawson, A. Dekusar, B. Delaney, P. Delosreyes, S. Derakhshandeh, A. Deshmukh, M. Desnoyers, O. E. Dial, M. DiDonato, S. Dikaleh, B. Dimock, T. Dixon, R. A. Donaton, B. Donovan, M. Doyle, Z. Dreiss, J. Drummond, E. A. Duch, N. Dunfee, L. Durst, R. A. Eades, N. Earnest-Noble, D. J. Egger, N. Einsidler, B. Ek, J. Ekman, S. Elrington, R. Elsasser, I. Elsayed, A. Emerich, S. Engelmann, K. R. Erickson, K. Escobar, A. Estes, M. Facchini, A. Falk, M. Farmer, B. Fearon, K. Ferguson, A. G. Fernandez, F. M. Fernandez, A. H. Ferrera, K. J. Ferris, B. Fett, A. Finck, J. Finley, L. E. Fischer, S. Flynn, F. F. Flöther, K. Fogel, J. P. Fouarge, G. Fraczak, P. Frank, D. Frank, J. C. Franquis, T. N. Friedhoff, D. Friedman, R. Frison, D. Fry, T. Fukui, B. Fuller, R. K. Fwamba, J. Gacon, M. Galante, J. M. Gambetta, C. A. Garcia, D. K. Garcia, J. M. Garcia, A. R. García, S. Garion, J. R. Garrison, B. P. Gaucher, A. Gebbie, J. Gerboth, S. Ghosh, F. Giancaspro, A. Giannetta, G. W. Gibson Jr., L. Gil, L. Gilgeous, G. Gionta, J. R. Glick, J. A. Glick, W. Glogowski, I. Goldman, J. T. Gomez, J. Gomez, S. de la Puente González, A. Gooding, R. T. Gordon, T. Greenfield, J. N. Greiner, L. Gruber, T. P. Gujarati, S. R. Gumede, K. Gunsky, H. Haas, R. Haight, S. A. Hall, I. Hamamura, J. B. Hannon, R. U. Haq, V. Harish, T. T. Harnack, S. Hart, G. S. Haspil, F. Haverkamp, V. Havlicek, C. Haymes, J. Haynes, A. He, M. B. Healy, R. Hebbar, J. Henspeter, F. J. C. Hernández, C. Herrick, J. Hertzberg, A. Hervier, H. Higgins, M. Hillenbrand, I. Hincks, C. W. P. Hoerst, K. L. Holland, B. Holt, M. Hopstaken, H. Horii, T. A. Horvath, R. House, D. Hrtanek, E. Huang, J. Huang, J. R. Huffman, L. Hung, T. Huynh, N. Ibrahim, T. Imamichi, B. Ingmanson, K. Inoue, J. Irish, S. Itoh, T. Itoko, K. Jadin, H. Jagannathan, K. Jambunathan, J. A. Janechek, A. Javadi-Abhari, M. J. Jeanson, T. A. Jennings, O. Jinka, T. Jochym-O'Connor, B. R. Johnson, C. Johnson, S. Johnston, M. R. Jokar, B. A. Jones, E. A. Joseph, P. Jurcevic, H. Kang, M. Karakas, A. Kariye, S. Kashiwa, Z. Kauffman, D. Kaulen, J. Kaus, K. Kawase, G. A. Keefe, P. Keefer, N. Keenan, J. Kern, A. Khokhar, H. Kim, S. Kim, T. Kiraly, S. Kirschner, S. Kister, K. Knodel, W. Kong, B. Kreeger, M. Kriegshauser, K. Krsulich, V. Kumar, M. Kumph, C. Kurotori, D. Kwee, M. LaDue, R. Lallement, W. F. Landers, O. Lanes, G. J. Lapeyre, C. Larsen, O. Lasisi, I. Lauer, D. Layden, S. LeClerc, K. Lee, C. Lee, S. Lekuch, R. Letzter, M. Li, I. Liepuoniute, P. Lindner, T. Lindquist, G. Lindsell, E. G. Liniger, V. Lipinska, C. B. Lirakis, J. Lishman, D. Lokken-Toyli, R. Lolowang, M. Lu, W. Lu, S. Lukashov, A. Lukin, J. Luna, Y. Luo, K. Main, R. Majumdar, M. Malekakhlagh, R. Malik, H. Mallela, H. Mamin, C. Mancini, P. Mann, C. Marroquin, Y. Martin, I. D. Martinez, J. J. Martinez, Y. G. Martín-Mantero, N. A. Masluk, M. Mastro, K. Masuda, A. Matsuo, T. Maurer, D. McClure, M. McDonald, S. McJunkin, D. McKay, P. McKeone, K. McLaughlin, B. McPeek, D. Meirom, F. J. Mertil, R. Mertz, S. S. Meshram, A. Miessen, D. Millar, J. Milloz, Z. Minev, A. Mitchell, B. Mitchell, A. Mitra, T. Mittal, S. Miyabe, H. Mohammadbagherpoor, T. Mori, J. G. Mosquera, S. Mostame, M. A. Mueed, S. Mukesh, T. Mulder, Y. Muranaka, W. Murphy, C. Murray, J. P. Mzila-O'Connor, T. Müller, J. Nah, V. K. Naik, K. Najafi, D. Nakano, Y. Nam, P. Nation, J. S. Newbury, C. Nirkhe, S. Nur, P. O'Brien, K. O'Connell, L. E. Ocola, R. Ohira, M. Ohtani, T. Onodera, E. Opata, J. Orcutt, L. Oscarlece, R. Otaolea, I. Othmani, J. A. Ott, G. I. Ovejero, R. Padbury, E. B. H. Padilla, S. Panda, J. M. Papalia, R. Paranjape,

P. Parazzoli, J. E. Park, B. Parney, R. T. Paske, D. Passarella, J. Patel, G. F. Paulik, E. Pednault, T. Pellegrini, B. Peropadre, K. S. Petrarca, S. Peyer, A. A. de la Peña, S. M. Pfaendler, M. Pfaffhauser, A. Phan, T. Phung, K. Pizzolato, B. Pokharel, T. Poole, K. Pooley, A. Popescu, C. Porter, A. Prabhakar, J. A. Prentice, E. Prew, E. Pritchett, M. Proissl, R. de Putter, D. Puzzuoli, A. Pyzyna, B. Quanz, E. Quigley, V. Radescu, J. J. Raftery, S. Raghunathan, A. Ragupathi, S. Rajalingam, P. Rall, J. Ralph, I. R. Ramallal, D. Ramirez, A. Rao, V. Rastunkov, A. Ray, S. Rayyan, C. T. Rettner, P. D. Reyes, C. Richard, D. Ristè, E. Rivera, P. Rivero, M. Rizzolo, J. M. Roberts, N. Robertson, N. Robinson-Duncan, K. Rodbell, D. J. Rodriguez, D. M. Rodriguez, V. Rodriguez-Toro, A. Rojas, A. Rose, K. Ross, M. Rossmannek, T. Roth, M. B. Rothwell, J. R. Rozen, J. J. Ruedinger, D. Rugar, J. A. Russo, L. Sabaloz, H. Saenz, M. Sagianis, M. Sandberg, S. Sanichar, A. Sasaki, R. Sathananthan, A. Sauerland, H. Saunders, F. Scafirimuto, C. Scerbo, P. Schardt, M. J. Scheckel, L. Schleeper, R. Schmidt, J. M. Schmoll, R. Schoenberger, T. L. Scholten, K. Schoneck, J. Schuhmacher, A. Seif, M. Selvanayagam, H. H. R. Sepulveda, I. F. Sertage, P. Shah, W. Shanks, D. Shao, V. Sharma, D. Sharma, O. Shehab, R. M. Shelby, S. Sheldon, J. Sherman, N. Shimada, O. Shtanko, S. A. Shurson, V. Siddhu, W. Simmons, N. Simon, E. N. Sirong, J. W. Sleight, J. Smith, J. A. Smolin, B. Snell, D. Soederstedt, J. G. Sogo, J. Speidell, W. Spratt, S. Spring, S. Srinivasan, A. A. Stabile, K. Stawiasz, M. Steffen, J. Stehlik, R. Steiner, D. Stephenson, C. Stuckey, L. V. Subramaniam, J. Summerour, Y. Sun, N. Sundaresan, K. Sung, J. Suttle, T. Syed, C. Tabachnick, F. Tacchino, M. Takeori, M. Takita, A. Talle, C. Tan, E. P. Tapia, J. Tarm, I. Tavernelli, N. Taylor, J. Tersoff, S. Thomas, T. Thorbeck, S. Thoss, K. Tian, A. Tiano, K. Tien, J. Timmerwilke, T. Timpane, M. Tokunari, R. Tolentino, M. Tolunay, G. Toth, G. Totir, J. Totte, K. D. Tran, M. Tran, M. Treinish, B. D. Trimm, C. Tuma, M. Turchetti, L. Turlapati, C. S. Tyberg, P. Tysowski, G. Ulsh, D. Underwood, F. Valenti, D. G. Valiñas, M. VanDieren, F. Varchon, R. Vaught, A. C. Vazquez, A. Vilches, V. Villar, D. Vinayagamurthy, M. H. Vishkasougheh, S. K. L. P. Vishwakarma, C. C. Voss, C. Vu, E. Vásquez, A. Wack, M. Walther, C. Wang, D. Wang, F. Wang, M. Ware, J. Watrous, B. Webb, P. Wendling, B. Werneke, N. White, S. Willenborg, J. Williams, D. Wisnieff, S. Woerner, A. Wolseth, R. Woo, K. Wood, S. P. Wood, C. J. Wood, J. R. Wootton, J. L. Wright, B. Wymore, L. Xing, C. Xiong, T. Yamamoto, H. W. Yan, J. Yau, J. Yeh, K. Yoagaraj, T. Yoder, T. Younger, J. Yu, V. Yuen, D. Zajac, D. Zarsky, B. Zazueta, G. Zettles IV, E. J. Zhang, H. Zhang, R. Zhao, G. Zhu, S. Zhuk, S. Ziemak, C. Zoufal. M.Z. and S.A. were supported by the U.S. Department of Energy, Office of Science, Basic Energy Sciences, under Early Career Award No. DE-SC0022716. Y.W. is supported by the RIKEN iTHEMS fellowship. This work used the Anvil supercomputer at Purdue University through allocation PHY220016 from the Advanced Cyberinfrastructure Coordination Ecosystem: Services & Support (ACCESS) programme, which is supported by National Science Foundation grant nos. 2138259, 2138286, 2138307, 2137603 and 2138296. This research used resources of the National Energy Research Scientific Computing Center, a DOE Office of Science User Facility supported by the Office of Science of the U.S. Department of Energy under contract no. DE-AC02-05CH11231 using NERSC award BES-ERCAP0024710. This research used the Lawrencium computational cluster resource provided by the IT Division at the Lawrence Berkeley National Laboratory (supported by the Director, Office of Science, Office of Basic Energy Sciences, of the U.S. Department of Energy under contract no. DE-AC02-05CH11231). We are indebted to Frank Pollmann and Sheng-Hsuan Lin for their contributions to the isoTNS code used in this work. S.A. thanks J. Hauschild for insightful conversations and support with the TeNPy library[33], which was used to implement the MPS simulations.

**Author contributions** Y.K. and A.E. ran hardware and numerical experiments. S.A., Y.W. and M.Z. developed the tensor network methods and ran the classical simulations. Y.K., K.X.W. and A.K. developed the device-calibration protocol and calibrated the quantum processor with S.R. and H.N. E.v.d.B. and K.T. developed the error-mitigation protocol. Y.K., A.E., S.A., M.Z., K.T. and A.K. analysed the data. All authors contributed to the manuscript. K.T. and A.K. designed the project. A.K. managed the project.

**Competing interests** The authors declare no competing interests.

**Additional information**
**Correspondence and requests for materials** should be addressed to Youngseok Kim, Andrew Eddins or Abhinav Kandala.
