## [Peer Review File · Nature]

Manuscript Title: Evidence for the utility of quantum computing before fault tolerance

Reviewer Comments & Author Rebuttals

Reviewer Reports on the Initial Version:

Referees' comments:

Referee #1 (Remarks to the Author):

This is experimental and theoretical work at the pinnacle of what is currently possible. It is clearly-described and technically correct. The results are described without hyperbole, with careful attention to what can and cannot be concluded from them. Indeed, the authors also give potential ways in which the classical approaches might be modified to close the gap, before concluding on a positive note that anticipated near-term improvements in quantum computations will likely extend it.

One minor comment: the initial $|0\rangle^{\otimes 127}$ state has finite energy density in the Ising model. The issue of whether evolution of this state has experienced the ETH-induced simplification of the dynamics exploited by DAOE[Ref] (and Phys. Rev. B 97, 035127 (2018) which features one of the present authors) is worthy of brief discussion. This will ultimately determine whether forthcoming experiments with larger operator support will indeed be beyond classical simulation.

Given this, it is with hesitation that I raise grounds upon which one might question whether this is a Nature-level advance. Similar claims for analogue quantum simulation have been made previously [for example in Nature Physics 8, 325 (2012)] – in the sense that experiments were carried out that extended beyond the capabilities of then current tensor network simulations. This work goes further, but the case for inclusion in Nature is marginal. I personally lean towards the positive – if only to encourage the type of clear and sober (whilst ultimately positive) assessment that this work presents.

One minor comment: the initial $|0\rangle^{\otimes 127}$ state has finite energy density in the Ising model. The issue of whether evolution of this state has experienced the ETH-induced simplification of the dynamics exploited by DAOE[Ref] (and Phys. Rev. B 97, 035127 (2018) which features one of the present authors) is worthy of brief discussion. This will ultimately determine whether forthcoming experiments with larger operator support will indeed be beyond classical simulation

Referee #2 (Remarks to the Author):

The aim of this paper is to demonstrate that a large noisy QPU can accurately estimate averages of

observables (here magnetic moments in a 2D transverse-field Ising model), agreeing with classical exact solutions when available, and extending this to qubit volumes where classical computers cannot compete due to lack of memory.

The paper implements advanced versions of quantum error mitigation (QEM) – noise models with linear and exponential zero-noise extrapolation (ZNE) - to successfully demonstrate the ability of existing devices to perform accurate computations at moderate qubit volumes that can be checked by brute force classical simulation. Moreover, it then extends the methods to larger qubit volumes (up to 127 qubits), far beyond brute force classical simulation (as well as beyond 1D and 2D tensor network methods).

The paper demonstrates a basic form of quantum advantage, namely ability to perform accurate calculation of expectation values for large QPUs, far beyond classical calculation due to classical memory limitations. Interestingly, there was effectively no quantum speed-up. Moreover, the circuit depths were quite shallow, so there is probably a long way to useful algorithms. But that is another matter. The present paper demonstrates quantum memory advantage for computing physical properties of emulated physical systems.

The experimental techniques, data quality, statistics etc. are of high quality and sound.

We recommend this paper to be published in Nature, basically in its present form.

Some comments:

Perhaps consider/quote?

Exponentially tighter bounds on limitations of quantum error mitigation
Yihui Quek, et al.; arXiv:2210.11505v2

p.2, top of left column:

The text around the $R_{ZZ}(-\pi/2)$ gate is perhaps too brief: at least add a reference to a paper showing how to reduce the full circuit with 2 CNOTs and 1q rotation gates to the 1 CNOT circuit displayed.

p.4, top of right column:

A reference to “light-cone and depth reduced (LCDR) circuits” might be appropriate. Bravyi and König, PRL 110, 170503 (2013) is an obvious but formal choice, maybe there are easier descriptive references.

Referee #3 (Remarks to the Author):

The authors report experimental results using a 127-qubit programmable IBM quantum computer. Relatively shallow quantum circuits are executed whose structure is suited to the connectivity limitations of the quantum device, yet could be used for simulating the time evolution of a non-

trivial, practically relevant quantum spin model. Comparable important works in the field -- many of which have been published in Nature -- can be categorised as follows: simulating random quantum circuits/boson sampling experiments; simulating fermionic models which include small-scale quantum chemistry, Hubbard model or “wormhole” experiments; and implementing small-scale quantum error correcting codes. The present work clearly distinguishes itself given its larger scale and its potential for useful applications of noisy quantum computers.

Further below are a series of comments about the paper but our view is that the paper does indeed deserve to be published in Nature.

The primary points supporting this view are:

- a. It demonstrates impressive hardware and software developments, i.e., the number of qubits and the number of applied gates is significantly larger than in prior experiments while non-trivial advances in error mitigation are demonstrated
- b. The implemented circuits may be of practical use in simulating quantum dynamics – this could be contrasted with prior large-scale random-circuit sampling demonstrations
- c. The authors make an effort to demonstrate that the implemented circuits cannot be simulated classically with reasonable amounts of resource

We suggest that the authors should address the following recommendations before a publication decision.

1. In the abstract the authors say: “a noisy 127-qubit processor can measure accurate expectation values for circuit volumes that are beyond brute-force classical computation”. This sentence should be rephrased, e.g., the term brute-force is ambiguous as some of the circuits are classically exactly simulable in the paper while tensor network techniques are not really brute-force

2. The inset in Fig. 2 (c) compares the RMS error of the PEC approach to the presently used ZNE and finds 50 orders of magnitude difference. This comparison can be misleading and should be revised. The PEC error estimates are based on theoretical estimation of the sampling cost using Hoeffding's inequality, and also assumptions on the error-mitigated estimator variance. This RMS error can be much higher than the experimental values in many instances, thus we don't think it is appropriate to compare this against the ZNE RMS error from the specific experiments. Furthermore, as the authors detail in the paper, the circuits are shallow and light-cone arguments guarantee a significantly reduced sampling cost which should be accounted for in the PEC estimates. It would still be misleading to compare upper bounds of PEC with actual errors of ZNE so perhaps best would be to remove the inset plot.

3. The largest tensor network simulations were run on 64 cores with 128GB RAM for only 30 hours. This is not fundamentally more than the quantum processor's wall time, yet classical computing resources are significantly cheaper. Significantly larger classical computers are available so it's natural for a reader to ask how much we could improve upon the tensor-network simulations by running them on a larger scale. There are no further comments on this in the main text except that the authors run the simulations with an increasing bond dimension and find it improves the precision a bit but not significantly. Although there's an estimate reported on what bond dimension would be needed for an exact simulation, the main text should comment on what resources one

would require for a classical approximation that achieves comparable precision to the experiment.

4. It will be valuable to both specialist and non-specialist readers alike if the authors are able to conclude by putting the present paper in context in terms of the progress that needs to be made in order to achieve practical quantum advantage.

5. We've been asked to comment on the appropriateness of the title and we do find that the title is suitable. The paper is indeed a rigorous effort to gather evidence that utility is achievable on pre-fault-tolerant quantum computing machines. This is a matter of great interest to the community.

Author Rebuttals to Initial Comments:

Referee #1 (Remarks to the Author):

This is experimental and theoretical work at the pinnacle of what is currently possible. It is clearly-described and technically correct. The results are described without hyperbole, with careful attention to what can and cannot be concluded from them. Indeed, the authors also give potential ways in which the classical approaches might be modified to close the gap, before concluding on a positive note that anticipated near-term improvements in quantum computations will likely extend it.

We thank the referee for their positive view of our work and tone of presentation.

One minor comment: the initial $|0\rangle^{\otimes 127}$ state has finite energy density in the Ising model. The issue of whether evolution of this state has experienced the ETH-induced simplification of the dynamics exploited by DAOE[Ref] (and Phys. Rev. B 97, 035127 (2018) which features one of the present authors) is worthy of brief discussion. This will ultimately determine whether forthcoming experiments with larger operator support will indeed be beyond classical simulation.

The referee brings up an interesting open question that we now address in our discussion of classical next steps, in the main text.

“While these methods[40, 41] can successfully capture the long-time dynamics of the low-weight observables of a 1D spin chain, their applicability to high-weight observables in 2D at intermediate times is not clear - particularly as these methods are explicitly constructed to truncate complex operators.”

Given this, it is with hesitation that I raise grounds upon which one might question whether this is a Nature-level advance. Similar claims for analogue quantum simulation have been made previously [for example in Nature Physics 8, 325 (2012)] – in the sense that experiments were carried out that extended beyond the capabilities of then current tensor network simulations. This work goes further, but the case for inclusion in Nature is marginal. I personally lean towards the positive – if only to encourage the type of clear and sober (whilst ultimately positive) assessment that this work presents.

We thank the referee for their recommendation to publish in Nature. We also note that while analog quantum simulators have indeed previously challenged classical tensor network methods, we believe that demonstrating this with a universal quantum computer at this scale, with the capability to address a wide range of problems, significantly expands the impact of our result.

Referee #2 (Remarks to the Author):

The aim of this paper is to demonstrate that a large noisy QPU can accurately estimate averages

of observables (here magnetic moments in a 2D transverse-field Ising model), agreeing with classical exact solutions when available, and extending this to qubit volumes where classical computers cannot compete due to lack of memory.

The paper implements advanced versions of quantum error mitigation (QEM) – noise models with linear and exponential zero-noise extrapolation (ZNE) - to successfully demonstrate the ability of existing devices to perform accurate computations at moderate qubit volumes that can be checked by brute force classical simulation. Moreover, it then extends the methods to larger qubit volumes (up to 127 qubits), far beyond brute force classical simulation (as well as beyond 1D and 2D tensor network methods).

The paper demonstrates a basic form of quantum advantage, namely ability to perform accurate calculation of expectation values for large QPUs, far beyond classical calculation due to classical memory limitations. Interestingly, there was effectively no quantum speed-up. Moreover, the circuit depths were quite shallow, so there is probably a long way to useful algorithms. But that is another matter. The present paper demonstrates quantum memory advantage for computing physical properties of emulated physical systems.

The experimental techniques, data quality, statistics etc. are of high quality and sound.

We recommend this paper to be published in Nature, basically in its present form.

We are grateful to the referee for their positive view of our manuscript, and the recommendation to publish.

Some comments:

Perhaps consider/quote?

Exponentially tighter bounds on limitations of quantum error mitigation
Yihui Quek, et al.; arXiv:2210.11505v2

We thank referee #2 for kindly suggesting this reference. Including the suggested references, we have included more references in similar context and add additional comment on supplementary material, SII.C:

“The two error mitigation protocols ZNE and PEC set out to improve expectation values within the allotted coherence time of the device. For these protocols, the hardware noise sets a constant coherence limit in both the depth and the number of qubits. A rough estimate [S13] for some generic device noise parameter λ indicates that one needs the product $nL\lambda$, for n - qubits and a circuit depth L , to be small. As such, these mitigation protocols are not expected to increase the circuit depth beyond what is permitted by the hardware constants. Recently derived information theoretic bounds, that are independent of the specific error mitigation protocols, provide formal support for this picture [S14–S16]. This indicates that going forward beyond the optimization of the protocols taking into account the actually

relevant circuit volume [S17], the central contribution for increased circuit volumes will be the improvement in hardware noise.”

[S14] Quek, Y., França, D. S., Khatri, S., Meyer, J. J. & Eisert, J. Exponentially tighter bounds on limitations of quantum

error mitigation (2022). URL <https://arxiv.org/abs/2210.11505>.

[S15] Tsubouchi, K., Sagawa, T. & Yoshioka, N. Universal cost bound of quantum error mitigation based on quantum estimation

theory. arXiv preprint arXiv:2208.09385 (2022).

[S16] Takagi, R., Endo, S., Minagawa, S. & Gu, M. Fundamental limits of quantum error mitigation. npj Quantum Information

8, 114 (2022). URL <https://doi.org/10.1038/s41534-022-00618-z>.

[S17] Tran, M. C., Sharma, K. & Temme, K. Locality and error mitigation of quantum circuits (2023). 2303.06496.

p.2, top of left column:

The text around the $R_{ZZ}(-\pi/2)$ gate is perhaps too brief: at least add a reference to a paper showing how to reduce the full circuit with 2 CNOTs and 1q rotation gates to the 1 CNOT circuit displayed.

We appreciate that the specialist reader may already be more familiar with the decomposition of an RZZ using two CNOT gates, but since we do not use the two-CNOT decomposition anywhere in the manuscript, we think that here the two-CNOT decomposition is less of a natural starting point than the one-CNOT decomposition that we present. Since the interested reader can trivially verify the decomposition we provide by multiplying the 4x4 matrices for each gate, we feel a derivation reducing from two CNOTs to one CNOT would be an unnecessary detour in the narrative.

To offer some more intuition, the cross resonance gate utilizes native RZX interaction to construct the CNOT gate (PhysRevB.81.134507), specifically RZX($\pi/2$) gate. From the known relationship between RZX($\pi/2$) -- CNOT and RZX($\pi/2$) -- RZZ($\pi/2$), one can deduce the relationship between CNOT – RZZ($-\pi/2$).

p.4, top of right column:

A reference to “light-cone and depth reduced (LCDR) circuits” might be appropriate.

Bravyi and König, PRL 110, 170503 (2013) is an obvious but formal choice, maybe there are easier descriptive references.

The restriction of simulation to a light cone is fairly common at this point, and there isn't a single representative citation that we are aware of that is appropriate. In fact, we are using LCDR in more specific context, where we discuss the light cone and circuit reduction for specific circuit and observable. The curious reader is instead encouraged to see the discussion in the supplementary section SIIV.

Referee #3 (Remarks to the Author):

The authors report experimental results using a 127-qubit programmable IBM quantum computer. Relatively shallow quantum circuits are executed whose structure is suited to the connectivity limitations of the quantum device, yet could be used for simulating the time evolution of a non-trivial, practically relevant quantum spin model. Comparable important works in the field -- many of which have been published in Nature -- can be categorised as follows: simulating random quantum circuits/boson sampling experiments; simulating fermionic models which include small-scale quantum chemistry, Hubbard model or “wormhole” experiments; and implementing small-scale quantum error correcting codes. The present work clearly distinguishes itself given its larger scale and its potential for useful applications of noisy quantum computers.

Further below are a series of comments about the paper but our view is that the paper does indeed deserve to be published in Nature.

The primary points supporting this view are:

- a. It demonstrates impressive hardware and software developments, i.e., the number of qubits and the number of applied gates is significantly larger than in prior experiments while non-trivial advances in error mitigation are demonstrated
- b. The implemented circuits may be of practical use in simulating quantum dynamics – this could be contrasted with prior large-scale random-circuit sampling demonstrations
- c. The authors make an effort to demonstrate that the implemented circuits cannot be simulated classically with reasonable amounts of resource

We thank the referee for their careful reading and understanding of our manuscript, and recommendation to publish in Nature.

We suggest that the authors should address the following recommendations before a publication decision.

1. In the abstract the authors say: “a noisy 127-qubit processor can measure accurate expectation values for circuit volumes that are beyond brute-force classical computation”. This sentence should be rephrased, e.g., the term brute-force is ambiguous as some of the circuits are classically exactly simulable in the paper while tensor network techniques are not really brute-force

The term “brute force” is now more clearly defined in e.g. the Methods section, as a naïve classical computation defined in contrast to more sophisticated tensor network approaches. In the abstract, we have now changed “We show that a noisy 127-qubit processor can measure accurate expectation values for circuit volumes that are beyond brute-force classical computation” to “We show that a noisy 127-qubit processor can measure accurate expectation values for circuit volumes at a scale beyond brute-force classical computation.”

2. The inset in Fig. 2 (c) compares the RMS error of the PEC approach to the presently used ZNE and finds 50 orders of magnitude difference. This comparison can be misleading and should be revised. The PEC error estimates are based on theoretical estimation of the sampling cost using Hoeffding's inequality, and also assumptions on the error-mitigated estimator variance. This RMS error can be much higher than the experimental values in many instances, thus we don't think it is appropriate to compare this against the ZNE RMS error from the specific experiments. Furthermore, as the authors detail in the paper, the circuits are shallow and lightcone arguments guarantee a significantly reduced sampling cost which should be accounted for in the PEC estimates. It would still be misleading to compare upper bounds of PEC with actual errors of ZNE so perhaps best would be to remove the inset plot.

We agree with the referee that this comparison was too complicated. We have simplified the discussion and plot to more clearly focus on the main point: the standard prediction for the sampling cost of using naïve PEC (i.e. without lightcone optimization) is astronomical for experiments at this scale on our hardware, motivating the study of other mitigation methods such as ZNE (or lightcone-PEC).

To streamline the main text, we have revised the relevant inset plot and moved it from the main text to the supplement, and also updated the main text and supplemental text. The main text now reads:

Notably, the number of samples employed here is dramatically smaller than the estimated sampling overhead for a naïve PEC implementation, see SIV.B. In principle, this disparity may be greatly reduced by more advanced PEC implementations using lightcone tracing [30], or by improvements in hardware error rates. As future hardware and software developments bring down sampling costs, PEC may be preferred when affordable to avoid the potentially biased nature of ZNE.

Previously the relevant plot compared the theoretically expected performance of a PEC-mitigated estimator against an average of many experiments considered to be instances of a ZNE-mitigated estimator; to avoid pitfalls in justifying this comparison, we removed ZNE from the plot, and show only γ^2 as an estimate of the PEC sampling cost, with no mention of RMS error anywhere. The comparison with ZNE in the supplement is reduced to one sentence in the supplemental text:

We see that ZNE performed well in the main-text experiments despite the presence of hardware noise predicted to make a naïve PEC implementation unaffordable, highlighting the importance of choosing an error mitigation scheme suitable for a given computational task and available quantum hardware.

And one sentence in the supplemental figure caption:

These astronomical values suggest that such a mitigation scheme would be untenable here, motivating the use of methods expected to have lower overhead such as ZNE or lightcone-optimized PEC.

Generally, we expect ZNE to have lower overhead as it need only mitigate errors in the circuit to the extent that they matter for the desired observable (much like lightcone-PEC ignores errors known to be inconsequential), whereas naïve PEC fully mitigates all errors in the circuit.

3. The largest tensor network simulations were run on 64 cores with 128GB RAM for only 30 hours. This is not fundamentally more than the quantum processor's wall time, yet classical computing resources are significantly cheaper. Significantly larger classical computers are available so it's natural for a reader to ask how much we could improve upon the tensor-network simulations by running them on a larger scale. There are no further comments on this in the main text except that the authors run the simulations with an increasing bond dimension and find it improves the precision a bit but not significantly. Although there's an estimate reported on what bond dimension would be needed for an exact simulation, the main text should comment on what resources one would require for a classical approximation that achieves comparable precision to the experiment.

Following the referee #3's suggestion, we have now expanded the following comment addressing required resources for a classical approximation to obtain exact solution for Fig. 4:

"The bond dimension needed to exactly represent the stabilizer state and its evolution to depth 20 at $\theta_h = \pi/2$ is 7.2×10^{16} , 13 orders of magnitude larger than what we considered; see SVIII. For reference, as the memory required to store an MPS scales as $\mathcal{O}(\chi^2)$, already a bond dimension of $\chi = 1 \times 10^8$ would require 400 PB, independent of any runtime considerations."

Further, in supplementary section SVIII, we estimate the bond dimension achievable on Summit with 250 PB of storage, assuming that one needed to store the MPS and then MPO-MPS environments for application of the two-qubit unitaries. This gives a maximum possible bond dimension of 2.5 million, well below the bond dimension needed to exactly represent the stabilizer state and its evolution to depth 20 at $\theta_h = \pi/2$.

4. It will be valuable to both specialist and non-specialist readers alike if the authors are able to conclude by putting the present paper in context in terms of the progress that needs to be made in order to achieve practical quantum advantage.

We thank the referee for this suggestion. The conclusion in the current text predicts that quantum advantage will emerge out of a competition of classical algorithms and noisy quantum circuits of increasing volumes driven by enhancements in hardware performance, with a likelihood that for a time improvements in one will stimulate counter-balancing

progress in the other. We feel further elaboration would be too speculative to merit inclusion to the present paper.

5. We've been asked to comment on the appropriateness of the title and we do find that the title is suitable. The paper is indeed a rigorous effort to gather evidence that utility is achievable on pre-fault-tolerant quantum computing machines. This is a matter of great interest to the community.

We thank the referee for their careful consideration and agreement on the title.